# Mortality in gastro-oesophageal reflux disease in a population-based nationwide cohort study of Swedish twins

Eivind Ness-Jensen [ID] ,[1,2,3] Giola Santoni,[1] Eivind Gottlieb-Vedi,[1] Anna Lindam,[1] Nancy Pedersen,[4] Jesper Lagergren [ID] [1,5]

For numbered affiliations see end of article.

**Correspondence to**
Dr Eivind Ness-Jensen, HUNT Research Centre, Department of Public Health and Nursing, NTNU, Norwegian University of Science and Technology, Levanger, Norway;
eivind.ness-jensen@ntnu.no

## ABSTRACT

**Objectives** The public health disorder gastro-oesophageal reflux disease (GORD) is linked with several comorbidities, including oesophageal adenocarcinoma (OAC), but whether life expectancy is reduced by GORD is uncertain. This study assessed all-cause and cancer-specific mortality in GORD after controlling for confounding by heredity and other factors.

**Design** Population-based cohort study from 1998 to 2015.

**Setting** Swedish nationwide study.

**Participants** Twins (n=40 961) born in 1958 or earlier in Sweden.

**Exposure** GORD symptoms reported in structured computer-assisted telephone interviews.

**Outcomes** The primary outcome was all-cause mortality and the secondary outcome was cancer-specific mortality among twins with GORD and twins without GORD. HRs and 95% CIs were analysed using parametric survival models, both in individual twin analyses and co-twin pair analyses, with adjustment for body mass index, smoking, education and comorbidity.

**Results** Among 40 961 individual twins, 5812 (14.2%) had GORD at baseline and 8062 (19.7%) died during follow-up of up to 16 years. The risks of all-cause mortality (HR=1.00, 95% CI: 0.94–1.07) and cancer-specific mortality (HR=0.99, 95% CI: 0.89–1.10) were not increased in individual twins with GORD compared with individual twins without GORD. Similarly, there were no differences in mortality outcomes in within-pair analyses. The OAC-specific mortality rate was 0.45 (95% CI: 0.32–0.66) per 1000 person-years in individual twins with GORD and 0.22 (95% CI: 0.18–0.27) per 1000 person-years without GORD, rendering an adjusted HR of 2.01 (95% CI: 1.35–2.98).

**Conclusions** GORD did not increase all-cause or cancer-specific mortality when taking heredity and other confounders into account. The increased relative risk of mortality in OAC was low in absolute numbers.

## INTRODUCTION

Gastro-oesophageal reflux disease (GORD) is defined by troublesome heartburn and acid regurgitation occurring at least weekly or GORD-specific complications.[1] GORD

### Strengths and limitations of this study

► The twin design that adjusts for heredity and shared familial confounders.
► The prospective and nationwide population-based approach that counteracts recall and selection bias, as well as chance errors.
► Valid and complete long-term follow-up using national registers.
► Assessment of potential confounders.
► No objective assessment of gastro-oesophageal reflux disease.

affects between 10% and 30% of adults in the Western world and is one of the most common reasons for visits to gastroenterologists and general practitioners.[2 3] Heredity, obesity and tobacco smoking are the only established risk factors, while socioeconomic factors (mainly educational level) might also influence the risk of GORD.[4–7] Twin studies have shown that the heritability for GORD is 31%–43%.[8 9] Because GORD is associated with several conditions, for example, cardiovascular disease, various gastrointestinal symptoms, anxiety, depression, sleep disorders,[10–13] reductions in health-related quality of life,[14 15] and oesophageal adenocarcinoma (OAC) and gastric cardia adenocarcinoma,[16] it has been hypothesised that GORD reduces life expectancy in general and increases mortality from cancer specifically. Considering the high prevalence of GORD, any influence on life expectancy would be important for healthcare and public health interventions. However, the research that has examined whether GORD increases the risk of mortality has been limited and provided conflicting results, some indicating a reduced survival and other not.[17–20] No previous study has taken the influence of all risk factors for

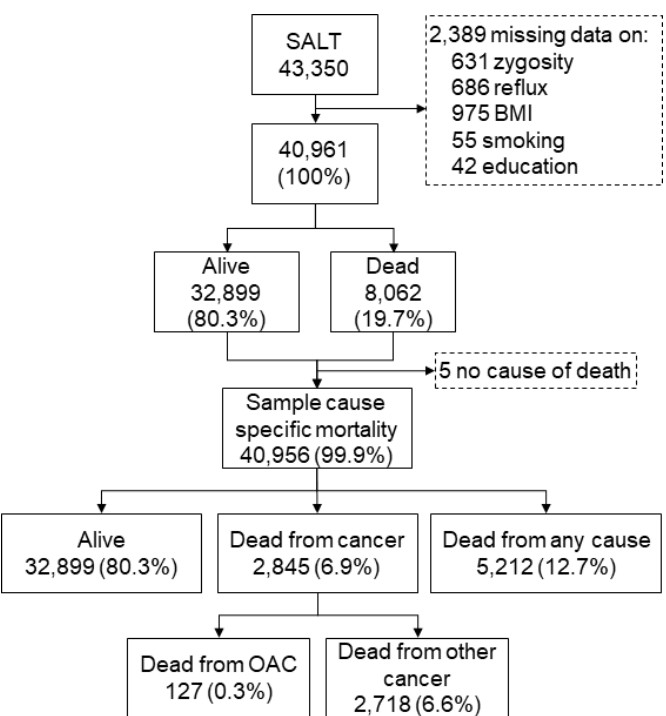

**Figure 1** Study population, sample, and vital status in twins with and without gastro-oesophageal reflux disease (GORD). BMI, body mass index; OAC, oesophageal adenocarcinom; SALT, screening across the lifespan twin cohort.

GORD into account as confounders, particularly not heredity or shared familial exposures.

The present study aimed to clarify whether GORD influences the mortality for all causes, cancer in general and OAC specifically by conducting a large and comprehensive twin study, controlling for genetic and familial influences, together with other potential confounders.

## METHODS
### Study design
This population-based twin study was based on data from the Swedish Twin Registry, during the study period 1998–2015. This Swedish Twin Registry incorporates comprehensive data retrieved directly from twins combined with data collected from Swedish national health registries. The personal identity number, which is assigned to each Swedish inhabitant, enabled exact linkage of participants' data between the data sources.[21]

### Patient and public involvement
Patients or the public were not involved in the design, or conduct, or reporting or dissemination plans of our research.

### Cohort
The study cohort was based on data from the Swedish Twin Registry, the largest and most comprehensive twin registry globally.[22 23] It was established in the late 1950s and includes virtually all twins born in Sweden from 1886 onwards. During 1998–2002, the Screening Across the

Lifespan Twin (SALT) study was performed with structured computer-assisted telephone interviews of twins born in 1958 or earlier and recorded in the Twin Registry, including assessment of GORD symptoms and risk factors for GORD.[22 23] Data from the SALT interviews were used to define the study cohort and to assess information about GORD and the potential confounders heredity, body mass index (BMI), tobacco smoking and education. Zygosity was assessed by a separate questionnaire sent to the twins. The twins were defined as monozygotic if both twins in a same-sexed pair reported they were 'alike as two peas in a pod' and as dizygotic if they reported to be 'not more alike than siblings'. This simple method has been shown to be 99% accurate in determining zygosity compared with DNA testing.[22] The Swedish Twin Registry is regularly updated with information from other nationwide Swedish registries, that is, the Cause of Death

**Table 1** Baseline characteristics of twins with and without GORD

| | GORD | | No GORD | |
|---|---|---|---|---|
| | Number (%) | | Number (%) | |
| Total | 5812 | (14.2) | 35149 | (85.8) |
| Age, years* | 56 | (41–95) | 56 | (41–99) |
| **Sex** | | | | |
| Men | 2673 | (46.0) | 16683 | (47.5) |
| Women | 3139 | (54.0) | 18466 | (52.5) |
| **Zygosity** | | | | |
| Monozygotic | 1444 | (24-8) | 8860 | (25.2) |
| Dizygotic | 4368 | (75.2) | 26289 | (75.8) |
| **BMI, kg/m²** | | | | |
| <25 | 2568 | (44.2) | 19577 | (55.7) |
| 25–30 | 2535 | (43.6) | 12909 | (36.7) |
| >30 | 709 | (12.2) | 2663 | (7.6) |
| **Tobacco smoking-status** | | | | |
| Never | 1236 | (21.3) | 8985 | (25.6) |
| Former | 3330 | (57.3) | 19104 | (54.4) |
| Current | 1246 | (21.4) | 7060 | (20.1) |
| **Education, years** | | | | |
| 0–9.5 | 3019 | (51.9) | 16405 | (46.7) |
| 9.5–12.5 | 1625 | (28.0) | 9895 | (28.2) |
| >12.5 | 1168 | (20.1) | 1168 | (25.2) |
| **Charlson Comorbidity Index** | | | | |
| 0 | 5190 | (89.3) | 31820 | (90.5) |
| 1 | 524 | (9.0) | 2883 | (8.2) |
| ≥2 | 98 | (1.7) | 446 | (1.3) |

*Median (range).
BMI, body mass index; GORD, gastro-oesophageal reflux disease.

**Table 2** Number of deaths and mortality rates for all-cause, cancer-specific and OAC-specific mortality in twins with and without GORD

| Outcome | Alive (n) | | Deaths (n) | | Mortality rates per 1000 person-years (95% CI) | |
|---|---|---|---|---|---|---|
| | GORD | No GORD | GORD | No GORD | GORD | No GORD |
| **All-cause mortality** | | | | | | |
| All twins | 6922 | 28 227 | 1140 | 4672 | 16.2 (15.3–17.2) | 16.2 (15.8–16.7) |
| Dizygotic* | 2091 | 2090 | 410 | 411 | 13.3 (12.1–14.7) | 13.3 (12.2–14.7) |
| Monozygotic* | 638 | 636 | 111 | 113 | 12.0 (10.0–14.4) | 12.2 (10.2–14.6) |
| **Overall cancer-specific mortality** | | | | | | |
| All twins | 5404 | 32 707 | 408 | 2437 | 5.8 (5.3–6.4) | 5.7 (5.5–6.0) |
| Dizygotic* | 2351 | 2351 | 150 | 150 | 4.9 (4.2–5.7) | 4.9 (4.2–5.7) |
| Monozygotic* | 701 | 706 | 48 | 43 | 5.2 (3.9–6.9) | 4.6 (3.5–6.3) |
| **OAC-specific mortality** | | | | | | |
| All twins | 5780 | 35 049 | 32 | 95 | 0.45 (0.32–0.66) | 0.22 (0.18–0.27) |
| Dizygotic* | 2489 | 2493 | 12 | 8 | 0.39 (0.23–0.74) | 0.26 (0.13–0.58) |
| Monozygotic* | 746 | 749 | 3 | 0 | 0.32 (0.10–1.58) | – |

*Discordant for GORD.
GORD, gastro-oesophageal reflux disease; OAC, oesophageal adenocarcinoma.

Registry, Cancer Registry and Patient Registry, which are briefly presented ahead.

*The Swedish Cause of Death Registry* provided data on all-cause and cancer-specific mortality. This registry includes the date of death and causes of death for all Swedish residents since 1961, regardless of whether they died in Sweden or abroad. The information about the date of death and cause of death is 100% and 99% complete, respectively.[24 25]

*The Swedish Cancer Registry* had information about the histological type of oesophageal cancer (adenocarcinoma). This registry started in 1958 and includes standardised records of all newly diagnosed malignancies in Sweden, including date of diagnosis, tumour site and histological type. Histological type is registered in accordance with the WHO's classification of histology (C24). The general completeness of the registry is 96% and it is 98% complete regarding the recording of OAC, and for these patients, the histological verification is 100% complete.[26 27]

*The Swedish Patient Registry* contained data on comorbidity. The registry contains date and International Classification of Diseases (ICD) versions 9 and 10 codes of diagnoses from all inpatient healthcare in Sweden from 1987 onwards and all specialist outpatient healthcare since 2001. This registry has a positive predictive value of any primary diagnosis close to 100%.[28] Diagnoses registered 3 years before and 3 years after the SALT interviews were included in the assessment of comorbidity. This restriction in time was done to counteract misclassification of comorbidity due to different lengths of follow-up among the participating twins.

## Exposure

The twins were defined as being exposed to GORD if they reported in the SALT interview to have: (1) heartburn at least weekly, (2) regurgitation at least weekly or (3) retrosternal pain at least weekly combined with antacid relief.[1]

## Outcomes

The main outcome was all-cause mortality, which included any deaths, regardless of cause. A secondary outcome was overall cancer-specific mortality, which included deaths related to any cancer (ICD-7 140–199 or ICD-10 C00-C97), excluding non-melanoma skin cancer (ICD-7 191 or ICD-10 C44). The other secondary outcome was OAC-specific mortality, defined as deaths related to OAC or gastro-oesophageal junctional adenocarcinoma (ICD-7 150 or 151.1 and C24 096 or ICD-10 C15 or C16.0 and C24 096).

## Confounders

Data on BMI, tobacco smoking and education were retrieved from the SALT interviews. BMI was calculated as the weight (kilograms) divided by the square height (metres). Smoking status included consumption of cigarettes, cigars and pipes. The level of education was assessed by the highest reported completed education qualification. Data on comorbidity were collected from the Swedish Patient Registry. The Royal College of Surgeons version of the Charlson Comorbidity Index was used to define and classify comorbidity.[29 30]

## Statistical analyses

Mortality rates per 1000 person-years were compared between individuals with and without GORD for all three

**Table 3** HR with 95% CI for all-cause, cancer-specific and OAC-specific-specific mortality in twins with and without gastro-oesophageal reflux disease

| Outcome | Model 1*<br>HR (95% CI) | Model 2†<br>HR (95% CI) | Model 3‡<br>HR (95% CI) |
|---|---|---|---|
| **All-cause mortality** | | | |
| All twins | 1.03 (0.97 to 1.10) | 1.03 (0.96 to 1.09) | 1.00 (0.94 to 1.07) |
| Dizygotic§ | 0.99 (0.87 to 1.13) | 1.04 (0.91 to 1.18) | 0.99 (0.87 to 1.14) |
| Monozygotic§ | 0.99 (0.79 to 1.24) | 1.05 (0.84 to 1.32) | 1.11 (0.87 to 1.40) |
| **Overall cancer-specific mortality** | | | |
| All twins | 1.04 (0.93 to 1.15) | 1.02 (0.92 to 1.14) | 0.99 (0.89 to 1.10) |
| Dizygotic§ | 1.00 (0.80 to 1.25) | 1.04 (0.83 to 1.30) | 0.99 (0.78 to 1.24) |
| Monozygotic§ | 1.13 (0.78 to 1.62) | 1.21 (0.84 to 1.75) | 1.28 (0.87 to 1.87) |
| **OAC-specific mortality** | | | |
| All twins | 2.09 (1.40 to 3.13) | 2.11 (1.41 to 3.15) | 2.01 (1.35 to 2.98) |
| Dizygotic§ | 1.50 (0.61 to 3.68) | 1.62 (0.70 to 3.78) | 1.44 (0.60 to 3.45) |
| Monozygotic§ | – | – | – |

*Adjusted for age and sex.
†Adjusted for age, sex, BMI, tobacco smoking status and education.
‡Adjusted for age, sex, BMI, tobacco smoking status, education and Charlson Comorbidity Index.
§Discordant for gastro-oesophageal reflux disease.
BMI, body mass index; OAC, oesophageal adenocarcinoma.

mortality outcomes. Parametric survival models with the Weibull distribution and sandwich estimator for the variance clustered by the twins' pair identity were used to calculate HRs with 95% CIs. These models correct for within twin pair dependency and help to avoid underestimation of the variance. The baseline hazard was modelled with both a linear and a quadratic time term to allow for more flexibility to the baseline function as the relationship between the baseline hazard and time was quadratic. The proportionality of the hazards was verified in all analyses. Time at risk was defined from the date of the SALT interview (1998–2002), that is, when GORD was assessed, until the date of death or the end of the study period (December 31, 2015).

The mortality among twins with GORD was compared with the mortality among twins without GORD in a stepwise series of analyses. First, external control analyses were performed using all individual twins, comparing individual twins with GORD with individual twins without GORD. Second, within-pair co-twin analyses of dizygotic twins discordant for GORD were performed. Third, within-pair co-twin analyses of monozygotic twins discordant for GORD were conducted. In the two latter analyses, only complete twin pairs were included. These three analysis steps were performed for each mortality outcome.

Stepwise adjustments for confounders were performed. First, a basic model adjusted for age (continuous) and sex. Second, the results were additionally adjusted for BMI (categorised into <25, 25–30 or >30), smoking (never, former or current) and years of completed education (0–9.5, 9.5–12.5 or >12.5 years). Third, the results were further adjusted for comorbidity (Charlson

Comorbidity Index score 0, 1 or ≥2),[29] which was done to assess whether comorbidity could explain any association between GORD and mortality.

In order to examine effect modification, analyses were stratified by age (≤60 or >60 years) and sex (except for the monozygotic twin analyses). In the monozygotic twin analysis of men aged 40–60 years, the HRs were estimated with exponential distribution and sandwich estimator for the variance, clustered by the twins' pair identity in order for the model to converge. This result should be similar to the model with the Weibull distribution, which did not converge in this analysis.

A senior biostatistician (GS) conducted data management and statistical analysis following a predefined study protocol. The statistical analyses were performed using Stata MP V.15 (StataCorp LP).

## RESULTS
### Participants
Among 43 350 individual twins who participated in SALT, 40 961 (95.5%) answered the questions relevant for the present study and were thus included in the final analysis. A flowchart describing the study cohort is shown in figure 1. Among the participating twins, 8062 (19.7%) died during follow-up of up to 16 years, including 2845 (6.9%) from any cancer and 127 (0.3%) from OAC. Characteristics of the included twins with and without GORD are shown in table 1. The median age was 56 years in both groups. In all, 14.2% had GORD and GORD was similarly common in both sexes and both dizygotic and monozygotic twins. Compared with twins without GORD,

**Table 4** HR* with 95% CI for all-cause, cancer-specific and OAC-specific mortality in twins with and without GORD

| | Age 40–60 years | | | Age >60 years | | |
| | Number of deaths | | | Number of deaths | | |
| Outcome | GORD | No GORD | HR (95% CI) | GORD | No GORD | HR (95% CI) |
| --- | --- | --- | --- | --- | --- | --- |
| Men | | | | | | |
| All-cause mortality | | | | | | |
| All twins | 140 | 766 | 0.97 (0.80 to1.17) | 437 | 2885 | 0.96 (0.87 to 1.06) |
| Dizygotic twins† | 58 | 54 | 0.96 (0.66 to1.39) | 152 | 164 | 0.91 (0.73 to 1.13) |
| Monozygotic twins† | 16 | 16 | 1.09 (0.57 to2.15) | 38 | 32 | 1.40 (0.92 to 2.13) |
| Overall cancer-specific mortality | | | | | | |
| All twins | 61 | 327 | 0.97 (0.73 to 1.29) | 151 | 971 | 0.97 (0.81 to 1.15) |
| Dizygotic twins† | 27 | 24 | 1.06 (0.60 to 1.87) | 51 | 57 | 0.80 (0.54 to 1.18) |
| Monozygotic twins† | 3 | 5 | 0.65 (0.12 to 3.38)‡ | 20 | 13 | 1.80 (0.96 to 3.38) |
| OAC-specific mortality | | | | | | |
| All twins | 14 | 22 | 3.71 (1.90 to 7.28) | 9 | 36 | 1.60 (0.77 to 3.32) |
| Dizygotic twins† | 6 | 1 | 2.07 (0.53 to 8.08) | 2 | 3 | 0.82 (0.15 to 4.61) |
| Monozygotic twins† | 1 | 0 | – | 2 | 0 | – |
| Women | | | | | | |
| All-cause mortality | | | | | | |
| All twins | 133 | 676 | 1.03 (0.85 to 1.26) | 430 | 2595 | 1.00 (0.90 to 1.11) |
| Dizygotic twins† | 52 | 56 | 1.03 (0.70 to 1.51) | 148 | 137 | 1.10 (0.87 to 1.40) |
| Monozygotic twins† | 15 | 20 | 0.75 (0.34 to 1.67) | 42 | 45 | 1.11 (0.78 to 1.57) |
| Overall cancer-specific mortality | | | | | | |
| All twins | 65 | 427 | 0.80 (0.61 to 1.05) | 131 | 712 | 1.10 (0.91 to 1.33) |
| Dizygotic twins† | 23 | 28 | 0.93 (0.52 to 1.65) | 49 | 41 | 1.30 (0.84 to 2.03) |
| Monozygotic twins† | 11 | 11 | 1.07 (0.41 to 2.77) | 14 | 14 | 1.06 (0.52 to 2.15) |
| OAC-specific mortality | | | | | | |
| All twins | 1 | 9 | 0.51 (0.06 to 4.09) | 8 | 28 | 1.81 (0.83 to3.94) |
| Dizygotic twins† | 0 | 0 | – | 3 | 2 | 1.39 (0.30 to 6.43) |
| Monozygotic twins† | 0 | 0 | – | 0 | 0 | – |

*Estimated with parametric survival model with Weibull distribution and sandwich estimator for the variance clustered by twins' pair ID, adjusted for BMI, tobacco smoking status, education, and Charlson Comorbidity Index.
†Discordant for GORD.
‡Estimated with exponential distribution and sandwich estimator for the variance clustered by twins' pair ID.
BMI, body mass index; GORD, gastro-oesophageal reflux disease; OAC, oesophageal adenocarcinoma.

the twins with GORD were more often overweight or obese, tobacco smokers, less educated and diagnosed with comorbidities (table 1). The study included 2501 dizygotic twin pairs discordant for GORD and 749 monozygotic twin pairs discordant for GORD.

### Mortality from any cause

The all-cause mortality rate of all individual twins was 16.2 (95% CI: 15.3–17.2) per 1000 person-years in twins with GORD and also 16.2 (95% CI: 15.8–16.7) per 1000 person-years in twins without GORD (table 2). In dizygotic twin pairs discordant for GORD, the all-cause mortality rates were 13.3 (95% CI: 12.1–14.7) per 1000 person-years in twins with GORD and 13.3 (95% CI: 12.2–14.7) per 1000 person-years for their co-twins without GORD. In monozygotic twin pairs discordant for GORD, the all-cause mortality rates were 12.0 (95% CI: 10.0–14.4) per 1000 person-years

in twins with GORD and 12.2 (95% CI: 10.2–14.6) per 1000 person-years in their co-twins without GORD.

The fully adjusted HR of all-cause mortality was 1.00 (95% CI: 0.94–1.07) comparing all individual twins with GORD with individual twins without GORD (table 3). In the dizygotic twin analysis, the corresponding HR was 0.99 (95% CI: 0.87–1.14). In the monozygotic twin analysis, the adjusted HR was 1.11 (95% CI: 0.87–1.40). The analyses stratified by sex and age showed similar HRs without any association between GORD and all-cause mortality (table 4).

### Mortality from any cancer

The overall cancer-specific mortality rate of all individual twins was 5.8 (95% CI: 5.3–6.4) per 1000 person-years in those with GORD and 5.7 (95% CI: 5.5–6.0) per 1000 person-years in those without GORD (table 2). The dizygotic twin analysis also showed similar cancer-specific

mortality rates in twins with GORD (4.9 (95% CI: 4.2–5.7) per 1000 person-years) and their co-twin without GORD (4.9 8 (95% CI: 4.2–5.7) per 1000 person-years). In the monozygotic twin analysis, the corresponding rates were 5.2 (95% CI: 3.9–6.9) per 1000 person-years in twins with GORD and 4.6 (95% CI: 3.5–6.3) per 1000 person-years in their co-twins with no GORD.

The fully adjusted HR of overall cancer-specific mortality was 0.99 (95% CI: 0.89–1.10) comparing all individual twins with GORD with individual twins without GORD (table 3). The corresponding HRs in dizygotic twins and monozygotic twins were 0.99 (95% CI: 0.78–1.24) and 1.28 (95% CI: 0.87–1.87), respectively. The analyses stratified by sex and age showed similar HRs and no association between GORD and overall cancer-specific mortality (table 4).

## Mortality from OAC
The OAC-specific mortality rate was 0.45 (95% CI:0.32–0.66) per 1000 person-years in all individual twins with GORD compared with 0.22 (95% CI: 0.18–0.27) per 1000 person-years in twins without GORD (table 2). In dizygotic twins, this rate was 0.39 (95% CI: 0.23–0.74) per 1000 person-years in the twins with GORD and 0.26 (95% CI: 0.13–0.58) per 1000 person-years in the twins without GORD. The mortality rate was 0.32 (95% CI: 0.10–1.58) per 1000 person-years in the monozygotic twins with GORD, whereas there was no OAC-specific mortality in the monozygotic twins without GORD.

The fully adjusted HR was 2.01 (95% CI: 1.35–2.98) for OAC-specific mortality comparing all individual twins with GORD with those without GORD (table 3). In dizygotic twins, the corresponding HR was 1.44 (95% CI: 0.60–3.45), whereas the statistical power was insufficient for monozygotic twin analysis. The HR was 3.71 (95% CI: 1.90–7.28) in men aged 40–60 years and 1.60 (95% CI: 0.77–3.32) in men aged >60 years (table 4). The stratified dizygotic twin analyses had low statistical power, but the fully adjusted HR for OAC-specific mortality was 2.07 (95% CI: 0.53–8.08) among men aged 40–60 years and 0.82 (95% CI: 0.15–4.61) among men aged >60 years.

## DISCUSSION
This large-scale twin study found no increased all-cause or cancer-specific mortality in twins with GORD compared with twins without GORD. The risk of mortality in OAC was higher in twins with GORD than in twins without GORD, but the absolute risk was still low.

Among methodological strengths is the twin design, which enabled the first study on the topic with adjustment for heredity and shared familial confounders. The prospective and nationwide population-based approach counteracted recall and selection bias, as well as chance errors. The high quality and complete data reduced misclassification and enabled long and complete follow-up of all participants. The assessment of mortality was valid and complete. The definition of GORD was the evidence-based Montreal consensus, which remains the definition of choice for

research purposes.[1] The prevalence of GORD in this study coincides well with the prevalence reported in similar Western populations,[2] indicating the validity of the definition of GORD. The assessment of potential confounders through the structured SALT interviews (BMI, tobacco smoking and education) and the Patient Registry (comorbidity) allowed for adjustment of all risk factors for GORD and mortality, that is, all plausible confounders. The rate of missing values for the variables included in the study was low, and all analyses were complete case analyses. The large sample size allowed for age-stratified and sex-stratified analyses to assess effect modification with age and sex.

There are also limitations. Some level of misclassification of GORD could not be avoided. Residual or unmeasured confounding cannot be ruled out in this observational study. The study lacks information on the medical and surgical treatment of GORD, so any change in mortality related to treatment could not be assessed. The dizygotic and monozygotic co-twin analyses had limited statistical power, although the results generally supported the overall findings.

The results of the present study showing no increased all-cause mortality in individuals with GORD corroborates the findings of our recent cohort study from Norway,[20] a cohort study from the USA[18] and a cohort study from Iran.[19] However, three cohort studies from the UK showed a 1.16-fold to 1.6-fold increase in mortality in people with GORD compared with the background population, the majority of deaths being due to cardiac disease.[17] The increased mortality found in some studies could be due to prevalent cancers provoking GORD symptoms. No earlier study has heredity as a confounder, although heredity is a strong risk factor for GORD.[8]

GORD is common in Western populations, with 10%–30% prevalence in adults.[2 3] The present study implies that individuals with GORD do not need to worry about any increased risk of dying. The increased risk of death from OAC should not be overemphasised because the absolute risk is still low even in the presence of GORD. However, if the incidence of OAC continues to increase strongly without any improvements in the survival, then the influence of mortality from this tumour could increase.

In conclusion, this nationwide Swedish population-based cohort study in twins with long and complete follow-up and adjustment for confounders indicates that GORD does not increase the risk of all-cause or cancer-specific mortality. Despite the increased relative risk of mortality from OAC in individuals with GORD, the absolute risk is still low.

**Author affiliations**
[1]Upper Gastrointestinal Research, Department of Molecular Medicine and Surgery, Karolinska Institute, Stockholm, Sweden
[2]HUNT Research Centre, Department of Public Health and Nursing, NTNU, Norwegian University of Science and Technology, Levanger, Norway
[3]Department of Medicine, Levanger Hospital, Nord-Trøndelag Hospital Trust, Levanger, Norway
[4]Department of Medical Epidemiology and Biostatistics, Karolinska Institute, Stockholm, Sweden
[5]School of Cancer and Pharmaceutical Sciences, King's College London, London, UK

**Acknowledgements** The authors are grateful to the Swedish Twin Registry for access to data. This Registry is managed by Karolinska Institutet and receives funding from the Swedish Research Council (grant 2017-00641).

**Contributors** EN-J: guarantor of the article; study concept and design; acquisition of data; analysis and interpretation of data; drafting of the manuscript; critical revision of the manuscript for important intellectual content; obtained funding. GS: study concept and design; analysis and interpretation of data; critical revision of the manuscript for important intellectual content; statistical analysis. EG-V, AL and NP: study concept and design; acquisition of data; analysis and interpretation of data; critical revision of the manuscript for important intellectual content. JL: study concept and design; analysis and interpretation of data; critical revision of the manuscript for important intellectual content; obtained funding. All authors approved the final version of the manuscript.

**Funding** This work was supported by the Swedish Research Council (SIMSAM) under grant number 2017-00641; Swedish Society of Medicine; and the United European Gastroenterology (UEG Research Prize to JL).

**Disclaimer** The funders did not have any role in the study design, in the collection, analysis or interpretation of data.

**Competing interests** None declared.

**Patient and public involvement** Patients and/or the public were not involved in the design, or conduct, or reporting, or dissemination plans of this research.

**Patient consent for publication** Not required.

**Ethics approval** The study was approved by the Regional Ethical Review Board in Stockholm (reference number 2010/582-31/1). All twins gave a broad informed consent for data collection and research when participating. The study protocol conforms to the ethical guidelines of the 1975 Declaration of Helsinki as reflected in a priori approval by the institution's human research committee.

**Provenance and peer review** Not commissioned; externally peer reviewed.

**Data availability statement** Data may be obtained from a third party and are not publicly available. The dataset used in this paper is available through application to The Swedish Twin Registry (https://ki.se/en/research/swedish-twin-registry-for-researchers).

**ORCID iDs**
Eivind Ness-Jensen http://orcid.org/0000-0001-6005-0729
Jesper Lagergren http://orcid.org/0000-0002-5143-5448

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
