## [Reviewer comments · BMJ Open]

ARTICLE DETAILS

TITLE (PROVISIONAL)	Mortality in gastro-oesophageal reflux disease in a population-based nationwide cohort study of Swedish twins
AUTHORS	Ness-Jensen, Eivind; Santoni, Giola; Gottlieb-Vedi, Eivind; Lindam, Anna; Pedersen, Nancy; Lagergren, Jesper

VERSION 1 – REVIEW

REVIEWER	Jonas Sanberg Ljungdahl Department of Surgery, Kolding hospital - a part of Lillebaelt Hospital, Denmark
REVIEW RETURNED	11-Mar-2020

GENERAL COMMENTS	Firstly, thank you for the opportunity to read an excellent and well-performed study. Using a combination of general registries and the twin registry, is both relevant and of high epidemiological value. Below is listed my concerns regarding the study, which warrants at least some explanation from the authors: Comorbidity The authors indicate, that Charlson Comorbidity Index is calculated and refers to Armitage et al 2010, that uses the RCS Charlson Score. Why have the authors chosen to use a surgical Charlson score instead of using the more generally applicable weights calculated by Quan et al. Am J Epidemiol. 2011;173(6):676–82? Potential confounders The reported weight and height of the study subjects are self-reported and despite going through the references to the twin registry, I cannot find indication of whether these parameters are validated through objective measurements? Education is grouped into years of education. Why not use groupings of the International Standard Classification of Education (ISCED) that reflects academic level of education and not simple years of education? https://ec.europa.eu/eurostat/statistics-explained/index.php/International_Standard_Classification_of_Education_(ISCED) Outcome The authors investigate patients who have indicated relevant GORD-symptoms and follow these patients until death or end of the follow-up period. However, to investigate cancer-specific mortality, it would be interesting to know more about the incidence of cancer in the study population. It is possible to survive oesophageal adenocarcinoma, in fact 5-year survival in population-based studies are 35-50%. Incidence of oesophageal adenocarcinoma and
--

	treatment is certainly relevant, if there are differences in tumour stage at presentation and subsequent possibility of curative treatment. Furthermore, and most importantly, treatment of GORD during the follow-up period, is extremely relevant for the outcomes and is not reported. Do the Swedish registries not allow for data on use of PPI and other anti-reflux drugs on an individual level? It certainly does allow for data on potential anti-reflux surgery. If the study subjects risk change during the study period due to treatment, the exposures effect on outcome may not be real. Minor typing or language errors identified: Manuscript Page 10, line 18: There seems to be a “help” missing between “and” and “to”. Page 10, line 25: First date of death? Each subject hopefully only has a single date of death? if not I would question the validity of the used registry. Figure Gastro-oesophageal reflux disease should be shortened GORD, not GOERD. In the explanation below the figure, a closing bracket is missing. I look forward to hearing the authors' reply to the abovementioned comments and thank the authors for an excellent manuscript.
--	--

REVIEWER	Tiffany Gill The University of Adelaide, Australia
REVIEW RETURNED	07-Apr-2020

GENERAL COMMENTS	Thank you for the opportunity to review this paper. My comments are as follows: Abstract: Clear Introduction: Clear and concise Methods: Well described. Statistical analysis: Why was both a linear and a quadratic time term used? Were any sensitivity analyses conducted? Results: Well described. Discussion: Are the authors able to suggest why there may be an increased relative risk of mortality from oesophageal adenocarcinoma (in spite of the low absolute risk)? Table 1 minor typing error in the table column heading Otherwise tables and figures are appropriate.
--

VERSION 1 – AUTHOR RESPONSE

Reviewer: 1
Reviewer Name: Jonas Sanberg Ljungdahl

Institution and Country: Department of Surgery, Kolding hospital - a part of Lillebaelt Hospital, Denmark

Please state any competing interests or state 'None declared': None declared

Firstly, thank you for the opportunity to read an excellent and well-performed study. Using a combination of general registries and the twin registry, is both relevant and of high epidemiological value. Below is listed my concerns regarding the study, which warrants at least some explanation from the authors:

Comorbidity

The authors indicate, that Charlson Comorbidity Index is calculated and refers to Armitage et al 2010, that uses the RCS Charlson Score. Why have the authors chosen to use a surgical Charlson score instead of using the more generally applicable weights calculated by Quan et al. Am J Epidemiol. 2011;173(6):676–82?

Authors' response: We have in fact recently conducted a systematic assessment of the Charlson Comorbidity Index in registry-based research, which included 16 studies (Brusselsaers et al. 2017;56:401-406 PubMed). In this assessment we found that the Charlson Comorbidity Index in the version of the Royal College of Surgeons (which we used in this study) is most up-to-date and easy-to-use, and is recommended in registry-based research. We have revised the Comorbidity paragraph accordingly: "The Royal College of Surgeons version of the Charlson comorbidity index was used to define and classify comorbidity. This is the recommended version for registry-based research."

Potential confounders

The reported weight and height of the study subjects are self-reported and despite going through the references to the twin registry, I cannot find indication of whether these parameters are validated through objective measurements?

Authors' response: The self-reported measures have not been validated through objective measurements.

Education is grouped into years of education. Why not use groupings of the International Standard Classification of Education (ISCED) that reflects academic level of education and not simple years of education? [https://eur01.safelinks.protection.outlook.com/?url=https%3A%2F%2Fec.europa.eu%2Furostat%2Fstatistics-explained%2Findex.php%2FInternational_Standard_Classification_of_Education_&data=02%7C01%7Ceivind.ness-jensen%40ki.se%7Cb3850a169f6a4a5f9d2808d7dae470d1%7Cbff7eef1cf4b4f32be3da1dda043c05d%7C0%7C0%7C637218547011272042&data=yxOAU4B%2BUyTjd8UtCnHwfiTgKgWleNCMPBiRoVyt7I%3D&reserved=0\(ISCED\)](https://eur01.safelinks.protection.outlook.com/?url=https%3A%2F%2Fec.europa.eu%2Furostat%2Fstatistics-explained%2Findex.php%2FInternational_Standard_Classification_of_Education_&data=02%7C01%7Ceivind.ness-jensen%40ki.se%7Cb3850a169f6a4a5f9d2808d7dae470d1%7Cbff7eef1cf4b4f32be3da1dda043c05d%7C0%7C0%7C637218547011272042&data=yxOAU4B%2BUyTjd8UtCnHwfiTgKgWleNCMPBiRoVyt7I%3D&reserved=0(ISCED))

Authors' response: We have chosen to group education into years of education according to the Swedish school system, which is the most commonly used and best available categorisation in studies from Sweden.

Outcome

The authors investigate patients who have indicated relevant GORD-symptoms and follow these patients until death or end of the follow-up period. However, to investigate cancer-specific mortality, it would be interesting to know more about the incidence of cancer in the study population. It is possible to survive oesophageal adenocarcinoma, in fact 5-year survival in population-based studies are 35-50%. Incidence of oesophageal adenocarcinoma and treatment is certainly relevant, if there are differences in tumour stage at presentation and subsequent possibility of curative treatment.

Authors' response: The median survival in oesophageal adenocarcinoma is less than 1 year, and the overall 5-year survival is <15% in Sweden. Yet, we agree that tumour stage and the following treatment depending on stage will influence the survival. However, it is unlikely that there would be any major differences between twins with GORD and without that could explain the findings.

Furthermore, and most importantly, treatment of GORD during the follow-up period, is extremely relevant for the outcomes and is not reported. Do the Swedish registries not allow for data on use of PPI and other anti-reflux drugs on an individual level? It certainly does allow for data on potential anti-

reflux surgery. If the study subjects risk change during the study period due to treatment, the exposures effect on outcome may not be real.

Authors' response: All patients (at least over 95%) with GORD receive treatment, so there is no range of exposure. As the Swedish Prescribed Drug Register only includes data since 1 July 2005, it is not possible to assess all medication used in this study from 1998. Data on anti-reflux surgery is available, but only about 5% of all patients with GORD are operated (Maret-Ouda et al. BMJ Open 2017;7:e016505) and in this cohort the number of operated patients will be too few to add any significance. In addition, we have recently shown that medical and surgical treatment of GORD are not associated with oesophageal adenocarcinoma risk (Lagergren Ann Surg. 2019 Nov 27. doi: 10.1097/SLA.0000000000003709. [Epub ahead of print]).

Minor typing or language errors identified:

Manuscript

Page 10, line 18: There seems to be a "help" missing between "and" and "to".

Page 10, line 25: First date of death? Each subject hopefully only has a single date of death? if not I would question the validity of the used registry.

Figure

Gastro-oesophageal reflux disease should be shortened GORD, not GOERD.

In the explanation below the figure, a closing bracket is missing.

Authors' response: Thank you for identifying these typos, which have been corrected in the revised manuscript.

I look forward to hearing the authors' reply to the abovementioned comments and thank the authors for an excellent manuscript.

Reviewer: 2

Reviewer Name: Tiffany Gill

Institution and Country: The University of Adelaide, Australia

Please state any competing interests or state 'None declared': None declared

Thank you for the opportunity to review this paper. My comments are as follows:

Abstract: Clear

Introduction: Clear and concise

Methods: Well described.

Statistical analysis: Why was both a linear and a quadratic time term used?

Authors' response: Because the relation between baseline hazard and time was quadratic, we included in the model all the terms of a quadratic function to allow more flexibility to the baseline function.

Were any sensitivity analyses conducted?

Authors' response: All analyses were conducted based on a pre-defined study protocol and no additional sensitivity analyses were conducted.

Results: Well described.

Discussion: Are the authors able to suggest why there may be an increased relative risk of mortality from oesophageal adenocarcinoma (in spite of the low absolute risk)?

Authors' response: GORD is the main established risk factor of oesophageal adenocarcinoma and the 5-year survival rate in oesophageal cancer is <15%, so the increased risk of mortality from oesophageal adenocarcinoma follows from this.

Table 1 minor typing error in the table column heading

Otherwise tables and figures are appropriate

Authors' response: Thank you for identifying the typo, which has been corrected in the revised manuscript.

VERSION 2 – REVIEW

REVIEWER	Jonas Sanberg Ljungdahl Department of Surgery, Kolding Hospital, a part of Lillebaelt Hospital Denmark
REVIEW RETURNED	24-Apr-2020

GENERAL COMMENTS	Thank you to the authors for their response on the reviewer-comments and revised manuscript. In general, the answers are adequate and clears up some misunderstandings about the paper. Although I recognize previous works by the authors, I do not agree with the added statement “This is the recommended version for registry-based research”. I would simply remove this sentence, as the RCS Charlson Index may be your preferred method, and your previous work indicate it should be more used, but it is not the most used or preferred method in general. With regards to oesophageal adenocarcinoma risk investigated in previous work by the group (Maret-Ouda et al. Ann Surg. 2019 Nov 27), I do agree that the previous work investigate surgical treatments possible associated with adenocarcinoma-risk. It does however not investigate medical treatment and employ some assumptions regarding obesity, smoking and other risk factors of adenocarcinoma. It is excellent work, but hardly relevant as a response to my question. There is a possible range of exposure based on length and effectiveness of medical GORD-treatment, but I recognise that these data are not available for analysis. There should be a single paragraph describing limitations of the study in the discussion. It is an excellent paper, but limitations should be at least mentioned. In all, I recommend the paper for publication with minor revisions.
---

REVIEWER	Tiffany Gill The University of Adelaide, Australia
REVIEW RETURNED	11-May-2020

GENERAL COMMENTS	I thank the authors for addressing the previous comments. Some minor issues with English Introduction: First paragraph, second last line should be “..has taken the influence..” Last paragraph, second line “..in specific..” should be “specifically” Methods - statistical analysis section “..and to help avoid ..” should be “and help to avoid..” My final comment is related to my previous question.
---

	"The baseline hazard was modelled with a linear and a quadratic time term" It would assist the reader if the reason provided in the response document was included in the text.
--	---

VERSION 2 – AUTHOR RESPONSE

Reviewer: 1

Reviewer Name: Jonas Sanberg Ljungdahl

Institution and Country: Department of Surgery, Kolding Hospital, a part of Lillebaelt Hospital, Denmark Please state any competing interests or state 'None declared': None declared.

Thank you to the authors for their response on the reviewer-comments and revised manuscript. In general, the answers are adequate and clears up some misunderstandings about the paper.

Although I recognize previous works by the authors, I do not agree with the added statement "This is the recommended version for registry-based research". I would simply remove this sentence, as the RCS Charlson Index may be your preferred method, and your previous work indicate it should be more used, but it is not the most used or preferred method in general.

Authors' response: The sentence has been removed from the manuscript.

With regards to oesophageal adenocarcinoma risk investigated in previous work by the group (Maret-Ouda et al. Ann Surg. 2019 Nov 27), I do agree that the previous work investigate surgical treatments possible associated with adenocarcinoma-risk. It does however not investigate medical treatment and employ some assumptions regarding obesity, smoking and other risk factors of adenocarcinoma. It is excellent work, but hardly relevant as a response to my question.

Authors' response: Our study in Annals of Surgery actually examined both surgical and medical treatment of GORD without finding any association with the risk of developing oesophageal adenocarcinoma. As data on medical GORD-treatment are not available and only about 5% of all GORD patients are operated for GORD, we have not assessed surgical treatment in this study. This is a limitation that we have added to the limitations paragraph of the revised discussion.

There is a possible range of exposure based on length and effectiveness of medical GORD-treatment, but I recognise that these data are not available for analysis.

Authors' response: We agree on this comment, but unfortunately medical GORD-treatment was not available. We have added this to the limitations paragraph of the revised discussion. It should be added that our recent study in Annals of Surgery did not find any association between medical GORD-treatment and risk of developing oesophageal adenocarcinoma.

There should be a single paragraph describing limitations of the study in the discussion. It is an excellent paper, but limitations should be at least mentioned.

Authors' response: We have added a separate paragraph highlighting the limitations of the study in the revised discussion section.

In all, I recommend the paper for publication with minor revisions.

Reviewer: 2

Reviewer Name: Tiffany Gill

Institution and Country: The University of Adelaide, Australia Please state any competing interests or state 'None declared': None declared

I thank the authors for addressing the previous comments.

Some minor issues with English

Introduction: First paragraph, second last line should be "..has taken the influence.."

Last paragraph, second line "..in specific.." should be "specifically"

Methods - statistical analysis section "..and to help avoid .." should be "and help to avoid.."

Authors' response: The English has been corrected.

My final comment is related to my previous question.

"The baseline hazard was modelled with a linear and a quadratic time term" It would assist the reader if the reason provided in the response document was included in the text.

Authors' response: The reason has been added to the manuscript.